# Electrochemical Reaction in Hydrogen Peroxide and Structural Change of Platinum Nanoparticle-Supported Carbon Nanowalls Grown Using Plasma-Enhanced Chemical Vapor Deposition

**Masakazu Tomatsu [1],\*, Mineo Hiramatsu [2], Hiroki Kondo [1], Kenji Ishikawa [1], Takayoshi Tsutsumi [1], Makoto Sekine [1] and Masaru Hori [1]**

[1] Graduate School of Engineering, Nagoya University, Nagoya 464-8603, Japan; hkondo@nagoya-u.jp (H.K.); ishikawa@plasma.engg.nagoya-u.ac.jp (K.I.); tsutsumi@plasma.engg.nagoya-u.ac.jp (T.T.); sekine@plasma.engg.nagoya-u.ac.jp (M.S.); hori@nuee.nagoya-u.ac.jp (M.H.)
[2] Graduate School of Science and Technology, Meijo University, Nagoya 468-8502, Japan; mnhrmt@meijo-u.ac.jp
\* Correspondence: tomatsu.masakazu@b.mbox.nagoya-u.ac.jp; Tel.: +81-52-789-3461

**Abstract:** Hydrogen peroxide ($H_2O_2$) reactions on platinum nanoparticle-decorated carbon nanowalls (Pt-CNWs) under potential applications were investigated on a platform of CNWs grown on carbon fiber paper (CFP) using plasma-enhanced chemical vapor deposition. Through repeated cyclic voltammetry (CV), measurements of 1000 cycles using the Pt-CNW electrodes in phosphate-buffered saline (PBS) solution with 240 μM of $H_2O_2$, the observed response peak currents of $H_2O_2$ reduction decreased with the number of cycles, which is attributed to decomposition of $H_2O_2$. After CV measurements for a total of 3000 cycles, the density and height of CNWs were reduced and their surface morphology changed. Energy-dispersive X-ray (EDX) compositional mapping revealed agglomeration of Pt nanoparticles around the top edges of CNWs. The degradation mechanism of Pt-CNWs under potential application with $H_2O_2$ is discussed by focusing on the behavior of OH radicals generated by the $H_2O_2$ reduction.

**Keywords:** plasma-enhanced chemical vapor deposition; carbon nanostructure; carbon nanowalls; metal nanoparticle catalyst; electrochemical

---

## 1. Introduction

Carbon nanowalls (CNWs) are a carbon nanomaterial composed of multilayered graphene vertically standing on a substrate. CNWs have a high aspect ratio over 100, sometimes over 1000, and a large surface area over 1.14 $m^2$/mg [1–3]. In the case of electrode applications in liquids, conventional carbon nanomaterials, including carbon black (CB) and carbon nanotube (CNT), are often used as composites with particles of catalytic metals, such as platinum (Pt), and stacked layers of those composites are formed. These structures often cause isolation between neighboring ones, which means disconnection of electron-conducting paths and dead spaces among the stacked enclosures where material gases and byproducts cannot go in and out. In fact, these problems are serious issues for in-liquid devices such as fuel cells and electrochemical sensors. In contrast, CNW electrodes hardly have such issues, since they have continuous and vertically aligned wall structures from bottom to top. In addition, CNWs will not aggregate even in liquids because of their self-supported wall structure. Furthermore, chemical terminations of CNW edges and surfaces can be changed easily and flexibly using post-growth treatments, such that their surface wettability can be controlled in a very wide

range from superhydrophilic to superhydrophobic [4]. Due to all of these features, CNWs are regarded as an ideal platform for electrochemical applications in liquids.

Hydrogen peroxide ($H_2O_2$) is a well-known strong oxidant for not only inorganic materials, such as carbon nanomaterials, metals, and semiconductors, but also organic ones including biomolecules. $H_2O_2$ is also often employed to remove unnecessary amorphous carbon components covering the surface of CNTs and other carbon nanomaterials for their purification. Moreover, $H_2O_2$ is used as the source gas of fuel cells [5]. In addition to these applications, $H_2O_2$ exists as reactive oxygen species in the human body and has an oxidative effect. $H_2O_2$ can induce oxidative damage in the human body and its storage causes several diseases [6]. Therefore, $H_2O_2$ is an important indicator of those diseases and has to be measured readily with a high degree of accuracy. For this purpose, various methods to detect $H_2O_2$ have been developed. Among these methods, liquid chromatography and colorimetry have high detection sensitivity, which can detect a minimum of 100 nM of $H_2O_2$ [7]. On the other hand, an electrochemical sensor is faster and easy to operate. However, its sensitivity is lower than that of the other methods. In the case of the electrochemical sensor, charge transfers during oxidation or reduction on the electrode surface are measured using a three-electrode system. An increased reaction field is highly recommended in order to reduce the required amount of sample, such as blood, and the burden on the subject. In general, Pt has been used as a working electrode for the detection of $H_2O_2$ [8–10]. In order to improve the detection range and limit, composites of Pt nanoparticles and various nanocarbon materials have been actively developed as a new electrode material with a large specific surface area [11–13]. Carbon nanomaterials have been widely used in both analytical and industrial electrochemistry owing to their low cost, wide potential window, and relatively inert electrochemistry. $H_2O_2$ is electroactive on catalytic metals such as Pt, although it is considered to be inactive on many typical carbon-based electrodes at room temperature. For example, when CNTs were used as a support material for Pt nanoparticles, a minimum detection of 10 nM was reported [14]. Similarly, Pt nanoparticle-decorated carbon nanowalls (Pt-CNWs) are also a promising candidate as a novel catalytic electrode with a very large specific surface area [14].

As mentioned above, $H_2O_2$ has often been used for the purpose of surface cleaning and functionalization for CNT, graphene, and nanodiamond [15–17]. For example, through $H_2O_2$ treatment at 70 °C, the amorphous carbon content can be removed from the CNT surface, and the crystallinity of CNT is improved. It is known that $H_2O_2$ can be decomposed on the Pt surface in the following reaction:

$$2H_2O_2 \rightarrow 2H_2O + O_2 \tag{1}$$

Another reaction can be considered as well:

$$H_2O_2 + e^- \rightarrow OH^- + OH\cdot \tag{2}$$

In this reaction, byproducts, such as OH radicals and $OH^-$ ions, possess very high chemical reactivity and can be strong oxidants. Several reports on $H_2O_2$ detection using Pt/carbon nanomaterials have been presented, while chemical reactions of $H_2O_2$ on those surfaces are not sufficiently clarified yet. In other words, secondary effects of the above byproducts on the surface of support materials, including amorphous carbon and graphene, have hardly been investigated in spite of the possible chemical reactivity of byproducts against carbon. The stability and lifespan of Pt-CNW electrodes immersed in solution containing $H_2O_2$ may be limited by the surface reaction related to the byproducts.

In the case of $H_2O_2$ detection using the electrochemical method where Pt/carbon nanomaterials are used as a working electrode, current peaks of oxidation and reduction reactions are evaluated as a function of electrode potential. Then, amperometric responses at various $H_2O_2$ concentrations are obtained at a fixed reduction potential by the periodic addition of small amounts of $H_2O_2$ solution. Here, amperometry is the conventional way to quantitatively monitor reduction or oxidation in the presence of a fixed potential, by measuring changes in current. It provides excellent limits of detection, good linear range, and reproducibility. In our previous study using Pt-CNW electrode,

it was confirmed that the morphology of Pt-CNWs hardly changed after repeated amperometry measurements up to 1.5 mM of $H_2O_2$ concentration more than 10 times [14]. On the other hand, a $H_2O_2$ treatment with an extremely high concentration of 9.8 M for a long period (4 days) induced morphological changes of CNW surfaces. Besides the sensor application, a significant decrease in the electrochemical surface area of Pt nanoparticles on CB as electrodes of polymer electrolyte fuel cells is observed after 5000–6000 start–stop cycles under applied voltage [18,19]. This decrease is related to the degradation of carbon-based support materials by the reaction with byproducts, resulting in aggregation of Pt nanoparticles. Reduction of the lifespan of electrodes due to the degradation of carbon-based support materials is a serious problem. The degradation or structure change of carbon-based support materials attributable to the $H_2O_2$ reaction on Pt surfaces depends on the $H_2O_2$ concentration and applied potential value as well as the crystallinity of the support materials. These dependencies should be investigated in detail, and the reaction processes and mechanism of $H_2O_2$ on the Pt/carbon nanomaterial surfaces have to be clarified for the practical use of CNWs as support material. In this study, structural changes of Pt-CNW electrodes after repeated cyclic voltammetry (CV) measurements with $H_2O_2$ up to 3000 cycles were investigated. The degradation mechanism of Pt-CNWs under potential application with $H_2O_2$ is discussed, focusing on the behavior of OH radicals generated by the $H_2O_2$ reduction.

## 2. Materials and Methods

### 2.1. Prepared CNWs

Carbon fiber paper (CFP; EC-TP1-060T, Toray Industries, Tokyo, Japan) with a fiber diameter of 7 μm was used as a substrate [20,21]. Because of its unique features, including high conductivity, chemical resistance, large surface area, and liquid permeability, CFP is promising as a base material of liquid-pervious electrode. CNWs were grown on the CFP by radical injection plasma enhanced chemical vapor deposition (RI-PECVD) using a $CH_4/H_2$ gas mixture [22]. The RI-PECVD system has a tandem structure consisting of two types of plasma sources. One is a parallel plate capacitively coupled plasma (CCP) source using very high frequency (VHF, 100 MHz) electric power. The other is a microwave (2.45 GHz)-excited surface wave plasma (SWP). The SWP source is mounted on the CCP source, and they are connected by a multirole electrode. $H_2$ gas with a flow rate of 50 standard cc per minutes (sccm) was introduced into the upper SWP region, and H atoms generated in the upper SWP source were injected into the lower CCP region through the multirole electrode. $CH_4$ gas with a flow rate of 100 sccm was introduced into the CCP region as a carbon source. Total gas pressure was maintained at 3 Pa. The microwave power of the $H_2$ SWP and the VHF power of the CCP were 400 and 200 W, respectively. Employing this tandem system, multiple radicals (carbon precursor and H atom) can be controlled independently. During the CNW growth, the CFP substrate was heated at 800 °C using a carbon heater. Growth experiments were carried out for 55 min. After CNW growth on the CFP, an Ar atmospheric pressure plasma treatment (NU-Rei Co.) at 6 kV was applied to the sample surfaces to make them hydrophilic. The surface of CNWs grown on CFP (CNWs/CFP) was decorated with Pt nanoparticles by a liquid phase reduction method. First, the CNWs/CFP was immersed in dihydrogen hexachloroplatinate (IV) hexahydrate ($H_2PtCl_6 \cdot 6H_2O$, 99.9%, Sigma-Aldrich, st. Louis, MO, United States of America). Then, by adding sodium tetrahydroborate ($NaBH_4$), Pt atoms were reduced and Pt nanoparticles were prepared on the CNW surface.

### 2.2. Electrochemical Measurements

Electrochemical measurements were performed using a standard three-electrode setup with a Solartron SI 1287 potentiostat (Solartron Analytical, Hampshire, UK). Three electrodes, a working electrode (Pt-CNWs), a reference electrode (Ag/AgCl), and a counter electrode (Pt rod), were connected to the potentiostat and immersed in the electrolyte. The electrolyte was phosphate-buffered saline (PBS, 15mL, 0.1M) with pH 7.4. In the CV measurements, the reactions of oxidation and reduction

were found as peak currents when the corresponding potentials were applied to the working electrode. The scanning range of potential was from –0.4 to 0.8 V at a sweep rate of 50 mVs$^{-1}$. $H_2O_2$ was used as a detection substance, and its concentration was adjusted to 240 μM in electrolyte. The cyclic test of the CV measurement was performed as follows. One cyclic measurement was defined as one cyclic sweep of potential, in which the potential was sequentially swept from 0.8 to –0.4 V and from –0.4 to 0.8 V. The potential was scanned for 1000 cycles in the electrolyte with 240 μM $H_2O_2$ as the first test. Then, the electrolyte was exchanged for the fresh one with the same $H_2O_2$ concentration (240 μM), and the electrode was cleaned up. The CNW electrode was gently submerged in pure water and slowly rinsed for about 1 minute. The potential was scanned again for 1000 cycles using the same electrode as the second test. This operation was carried out once again to measure up to 3000 cycles (third test) in total (3000-cycle test).

*2.3. Solid Analysis*

After the 3000-cycle test, the Pt-CNWs were evaluated using scanning electron microscopy (SEM), energy-dispersive X-ray spectroscopy (EDX), and Raman spectroscopy. SEM images were taken using semi-in lens SEM equipment (SU8230, Hitachi, Tokyo, Japan) with an accelerating voltage of 10 kV. The EDX mapping was done using an energy-dispersive x-ray analyzer (X-Max50, Oxford Instruments, Abingdon-on-Thames, British) at an accelerating voltage of 15 kV. Raman spectra were measured by Raman microscope (inVia confocal Raman microscope, Renishaw, Wotton-under-Edge, British). Raman measurement was conducted using an Nd:YAG laser (532 nm) as an excitation light. The solutions in the electrochemical measurement cells before and after the measurements were analyzed by high-performance liquid chromatography (HPLC; Prominence, Shimadzu, Kyoto, Japan).

## 3. Results

*3.1. Observation of Shape of CNWs After Cycle Test*

Figure 1 shows the SEM images of Pt-decorated CNWs grown on carbon fiber before (a–c) and after (d–f) the 3000-cycle test; a,d show plane-view images, and b,c,e,f show cross-sectional view images. Figure 1c,f are magnified images corresponding to the areas indicated by dotted squares in Figure 1b,e, respectively. Figure 1a,d show that the density of walls on the CFP decreased after the 3000-cycle test. Here, the density of CNWs was evaluated as the estimated net length of the walls in μm$^2$ area [23]. The density of CNWs decreased from 3.9 to 2.5 μm/μm$^2$ after the 3000-cycle test, and their height decreased by 500 nm, as shown Figure 1b,e. These kinds of structural changes were not observed for CNWs without Pt nanoparticles. Therefore, these structural changes are attributed to any reactions related to the Pt surface. In Figure 1c, small bright spots distributed uniformly on the CNW surfaces indicate the uniform distribution of Pt nanoparticles. In contrast, sparse and larger bright spots seen in Figure 1f indicate the agglomeration of Pt nanoparticles after the 3000-cycle test.

*3.2. Observation of Pt NP Supported Form*

Figure 2 shows SEM images of Pt-decorated CNWs (2a,d), EDX elemental mapping of C signal (2b,e) and Pt signal (2c,f), and line profiles of Pt/C ratios (2g) before and after the 3000-cycle test. As shown in Figure 2f, the Pt nanoparticles were agglomerated around the top edge region after the test, while C signal distributions were homogeneous either before or after the test (Figure 2b,e). These kinds of structural changes were not observed for CNWs without Pt nanoparticles. Line profiles of Pt/C compositional ratios before and after the 3000-cycle tests shown in Figure 2g were obtained from the dotted rectangle regions in Figure 2a–f. The x-axis in Figure 2g is a distance from the root of CNWs to the edge. A sharp peak found after the 3000-cycle test indicates the segregation of Pt around the top edge region of CNWs, while the Pt distribution on the CNWs shows only a very broad peak before the test. This result suggests migration and aggregation of Pt nanoparticles during the electrochemical measurements in $H_2O_2$.

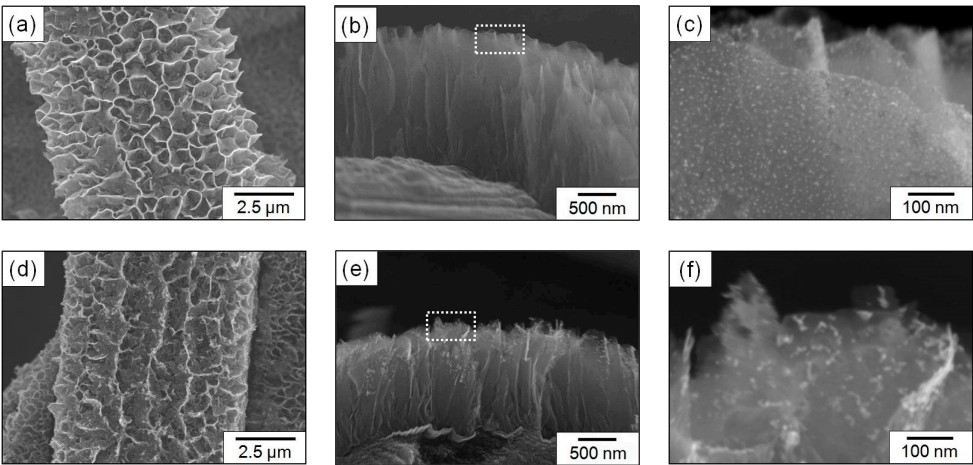

**Figure 1.** SEM images of (**a**) top view, (**b**) cross-sectional view, and (**c**) magnified view of edges of Pt-decorated carbon nanowalls (CNWs) grown on carbon fiber. SEM images of (**d**) top view, (**e**) cross-sectional view, and (**f**) magnified view of Pt-decorated CNWs grown on carbon fiber after 3000 cycles of cyclic voltammetry (CV) measurements in $H_2O_2$.

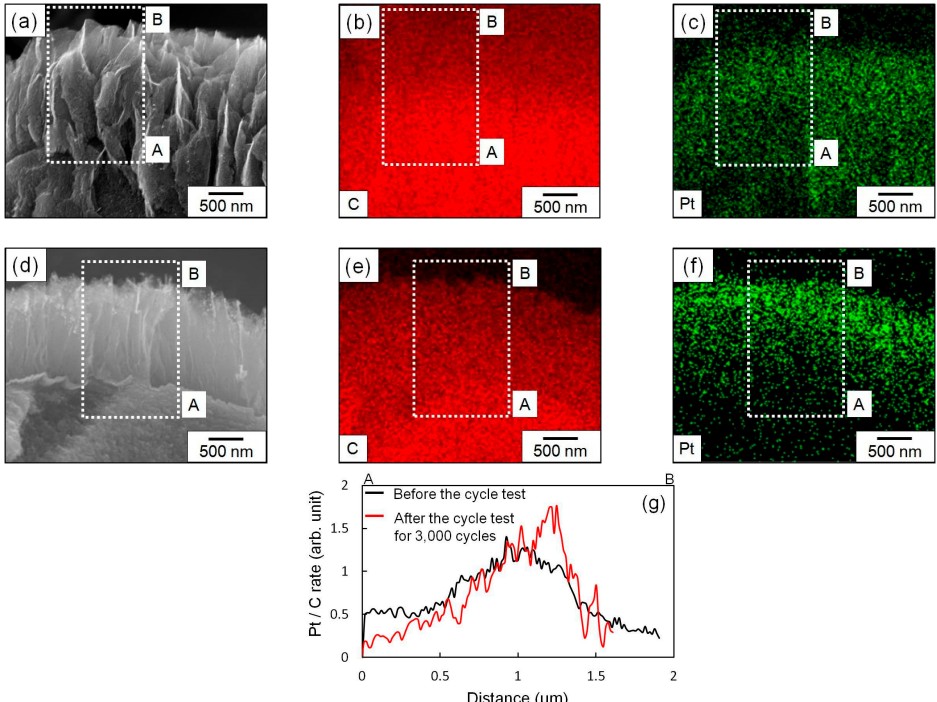

**Figure 2.** (**a**) SEM image of cross-sectional view of Pt-decorated CNWs grown on carbon fiber. EDX elemental mappings of C and Pt in Pt-CNWs: (**b**) cross-sectional image of C signal and (**c**) cross-sectional image of Pt signal, corresponding to the SEM image shown in Figure 2a. (**d**) SEM image of cross-sectional view of Pt-decorated CNWs grown on carbon fiber after 3000 cycles of CV measurements in $H_2O_2$. EDX elemental mappings of C and Pt in Pt-CNWs after 3000 cycles of CV measurements in $H_2O_2$: (**e**) cross-sectional image of C signal and (**f**) cross-sectional image of the Pt signal, corresponding to the SEM image shown in Figure 2d. (**g**) EDX line profiles of Pt to C signals in Pt-decorated CNWs before and after 3000 cycles of CV measurements in $H_2O_2$, measured along the A–B line shown in a–f.

### 3.3. Signal Analysis of CV Measurements

Figure 3a shows cyclic voltammogram responses for the 20th, 500th, and 1000th scan of the first test of 1000 cycles using the Pt-CNW electrode in PBS solution with 240 µM of $H_2O_2$. A CV curve for the

first scan without $H_2O_2$ is also indicated for comparison. Regardless of the number of CV measurement cycles, reduction peaks were found to be negative at a potential of –0.1 V. Because this kind of response peak current was not observed in the case without Pt nanoparticles in Figure 3b, the reduction peaks at –0.1 V are attributed to the catalytic reactions on the Pt nanoparticle surface. A similar peak was also found in the case without $H_2O_2$. This is attributed to dissolved oxygen molecular gas. As the number of cycles increased from 20 to 1000, the reduction peak currents at a potential of –0.1 V decreased. This suggests the occurrence of decomposition of $H_2O_2$. At the 500th cycle, new, small reduction and oxidation peaks appeared at potentials of 0.05 and 0.2 V, respectively. These peaks became sharper at the 1000th cycle; they were not observed at the 20th cycle or without $H_2O_2$. These results indicate the change of solution components and surface conditions of electrode with the electrochemical reaction proceeding. Figure 3c shows $H_2O_2$ reduction current peaks at a potential of –0.1 V of CV response as a function of number of cycles. The *x*- and *y*-axes in Figure 3c are potential (V vs. Ag/AgCl) and response current (mA), respectively. In this experiment, the Pt-CNW electrode was cleaned up and a new PBS solution with 240 μM of $H_2O_2$ was introduced into the measurement cell before the first, second, and third tests of 1000 cycles. The response currents decreased with an increased number of cycles in all sets of 1000 cycles. Decreased peak current during each set of 1000 cycles was almost constant at about 13%, probably due to the decomposition of $H_2O_2$ as a result of the electrochemical measurements using the Pt-loaded electrode. Actually, $H_2O_2$ concentration measured by HPLC decreased by about 60 μM after 20 cycles of CV measurement. On the other hand, the decrease of peak current became saturated around 900, 700, and 900 cycles in the first, second, and third tests of 1000 cycles, respectively. This saturation may be due to the change of surface condition of the Pt-CNW electrode. For example, the amorphous carbon component on the CNW surface could be removed by the $H_2O_2$ treatment [24]. Removal of the amorphous carbon component would improve the electrical contact between the Pt and CNW electrode. As mentioned before, since the PBS solution was replaced with a new solution 240 μM $H_2O_2$ and the Pt-CNW electrodes were cleaned up after the first and second tests, the response currents jumped somewhat at the start of the second and third tests. However, these current recoveries did not reach the original value, probably due to the decreased active surface area of Pt nanoparticles as a result of Pt agglomeration, as shown in Figure 1f.

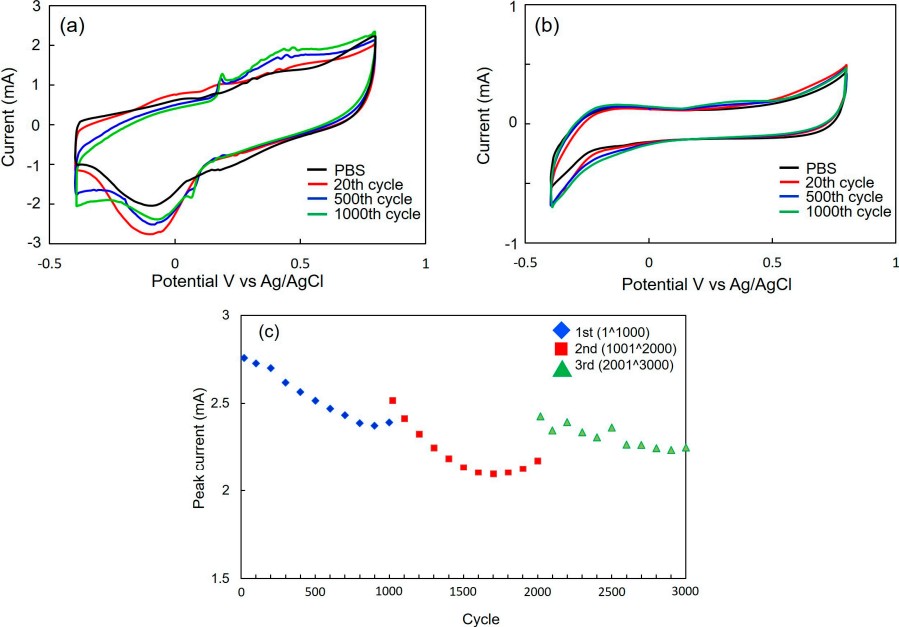

**Figure 3.** Cyclic voltammogram responses for the 20th, 500th, and 1000th scans of first test of 1000 cycles using (**a**) the Pt-CNW electrode and (**b**) the CNW electrode without Pt nanoparticles in PBS solution with 240 μM of $H_2O_2$, together with a CV curve for the first scan without $H_2O_2$. (**c**) Variation of peak current value at the potential of –0.1 of CV curves recorded from 1 to 3000 cycles.

### 3.4. Investigating Factors of Structural Change

Figure 4a,b show SEM images of as-formed Pt-CNWs and Pt-CNWs after the 3000-cycle test in PBS solution without $H_2O_2$. Figure 4c,d show SEM images of Pt-CNWs after immersion in PBS solution with and without 240 μM of $H_2O_2$ for 42 hours, which is the same period required for the electrochemical measurements as the 3000-cycle test. Comparing these images, structural change was hardly found, although the CNW densities were slightly different at 3.9, 4.0, 3.4, and 3.5 μm/μm². These results indicate that the significant change of Pt-CNWs found after the cyclic test could only occur under the potential application with $H_2O_2$.

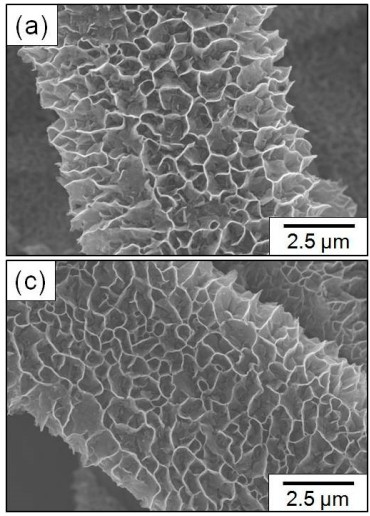
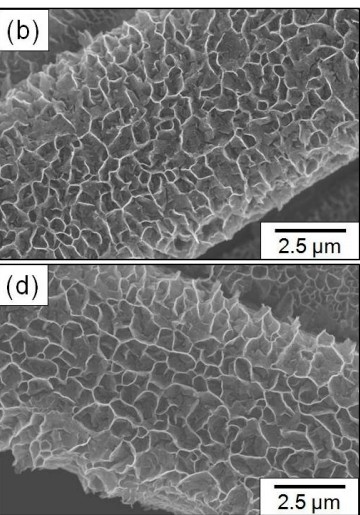

**Figure 4.** Top-view SEM images of (**a**) as-formed Pt-CNWs, (**b**) Pt-CNWs after the 3000-cycle test in PBS solution without $H_2O_2$, (**c**) Pt-CNWs after immersion in PBS solution with 240 μM of $H_2O_2$ for 42 hours, and (**d**) Pt-CNWs after immersion in PBS solution without $H_2O_2$.

### 3.5. Analysis of the Crystal Structure of CNWs

Figure 5 shows Raman spectra of as-formed Pt-CNWs and Pt-CNWs after the 3000-cycle test in PBS solution with and without 240 μM of $H_2O_2$. Raman spectra of Pt-CNWs just after immersion in PBS with and without $H_2O_2$ are also shown in the figure, together with that of CFP without CNWs for comparison. In this figure, all spectra are normalized to the intensities of G-band peaks in their respective spectra. Three typical peaks, G-band, D-band, and D'-band peaks, were found in all spectra of CNWs. The G-band peak at 1580 cm$^{-1}$ results from the six-membered ring structure of graphite. The D-band peak at 1350 cm$^{-1}$ indicates the disorders of six-membered ring structures [25]. The D'-band peak at 1620 cm$^{-1}$, appearing as a shoulder of the G-band peak, is related to finite sizes of graphite crystallites and graphene edges [26]. Relatively large and sharp D-band and D'-band peaks are typical of CNWs. These intensities are much smaller than that for the graphite (CFP) without CNWs. Intensity ratios of D- and G-band peaks ($I_D/I_G$) were obtained using the area intensities of respective peaks. The $I_D/I_G$ ratio is often used to evaluate the densities of defects or disorders in carbon structures [27,28]. The $I_{D'}/I_G$ ratios were also calculated using the area intensities of D'- and G-band peaks, which would indicate the amount of graphene edges. The $I_D/I_G$ and $I_{D'}/I_G$ ratios of Raman spectra shown in Figure 5 are summarized in Table 1. After the 3000-cycle test with $H_2O_2$, both $I_D/I_G$ and $I_{D'}/I_G$ ratios in the Raman spectra of Pt-CNWs drastically decreased. This result is consistent with the fact that etching of the CNWs was found in the SEM images after the cyclic test with $H_2O_2$. Almost no change in the width of all peaks also indicates the etching of CNWs simply from the top edges without any structural degradation, since the peak width also generally means a fluctuation of crystalline structure (data for peak width are not shown). On the other hand, the $I_D/I_G$ ratio increased by 3000 cycles with PBS without $H_2O_2$, and also increased due to immersion in PBS

with and without $H_2O_2$. No significant difference arose between whether the potential was applied or not. One possible reason for such increases in $I_D/I_G$ ratio is the intercalation of ions between graphene layers, because the PBS solution includes light metals, such as sodium and potassium [29,30].

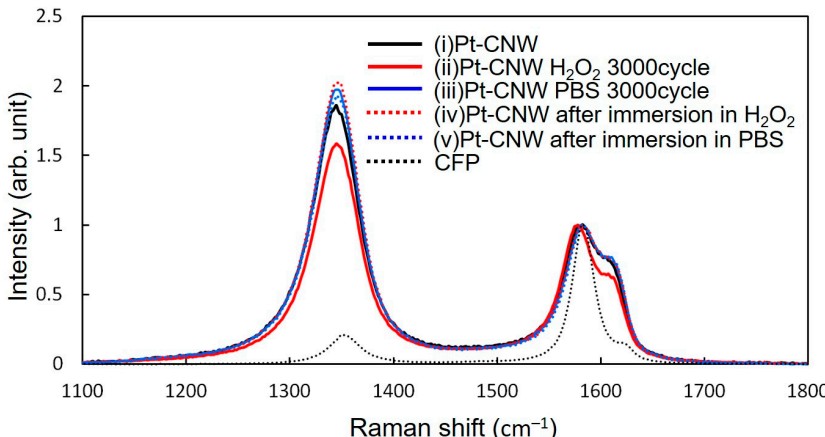

**Figure 5.** Raman spectra of as-formed Pt-CNWs and Pt-CNWs after 3000-cycle test in PBS solution with and without 240 μM of $H_2O_2$, with those of Pt-CNWs just after immersion in PBS with and without $H_2O_2$ for 42 hours, and that of CFP without CNWs.

**Table 1.** Ratio of $I_D/I_G$ and $I_{D'}/I_G$ in each condition of (i)–(vi).

| | Pt-CNWs | Pt-CNWs $H_2O_2$ 3000 Cycles | Pt-CNWs PBS 3000 Cycles | Pt-CNWs after Immersion $H_2O_2$ | Pt-CNWs after Immersion PBS | CFP |
|---|---|---|---|---|---|---|
| $I_D/I_G$ | 2.12 | 1.83 | 2.32 | 2.38 | 2.24 | 0.29 |
| $I_{D'}/I_G$ | 0.13 | 0.1 | 0.13 | 0.13 | 0.13 | 0.05 |

## 4. Discussion

As the results in Figures 1–5 show, it was found that Pt-CNWs were drastically etched from their top edges only under potential application with $H_2O_2$. In general, the reduction reaction of $H_2O_2$ depends on the solution conditions. In an acid solution, $H_2O_2$ is decomposed into $H_2O$ and $O_2$ [31]. On the other hand, it is known that anions ($OH^-$) are produced in neutral or alkaline solutions [32,33]. Because the PBS solution used in this experiment was maintained to be neutral, it is deduced that $H_2O_2$ was decomposed into $OH^-$ ions by the reduction reaction (2) on the surface of Pt nanoparticles. Provided that $OH^-$ ions are generated, pH near the electrode surface may shift to be alkaline. Production of OH radicals in acidic or alkaline solutions has been reported [34]. For example, OH radicals were produced from $H_2O_2$ in the presence of dissolved oxygen via electrochemical reaction [35]. In addition, it has been reported that graphene was corroded due to the action of OH radical [36]. Furthermore, it is also known that carbon materials, such as CNTs in contact with catalytic Pt nanoparticles, were etched in ambient oxygen [37]. Figure 6 depicts the degradation model of Pt-CNWs via the secondary reaction with byproducts of $H_2O_2$ reduction. Graphene edges have a higher chemical reactivity than graphene basal planes [38–40]. Therefore, etching or corrosion of CNWs via the byproducts of $H_2O_2$ reduction reaction occurred mainly from their top edges, resulting in the agglomeration of Pt nanoparticles at the top region. Based on these considerations, it is deduced that the secondary reaction by the OH radicals at the Pt-CNW surfaces can be induced only under potential application with $H_2O_2$. Then, they cause structural degradation of the Pt-CNWs and decreasing response peak current of $H_2O_2$ in the CV measurements. This kind of corrosion will be more likely to occur on the surface of amorphous carbon, which explains the significant agglomeration of Pt nanoparticles on the CB. The findings obtained in this study will open the way to designing a state-of-the-art electrode with high stability and durability for electrochemical application, as well as

to realize selective modification, structural control, and patterning of CNWs for the future in-liquid device applications.

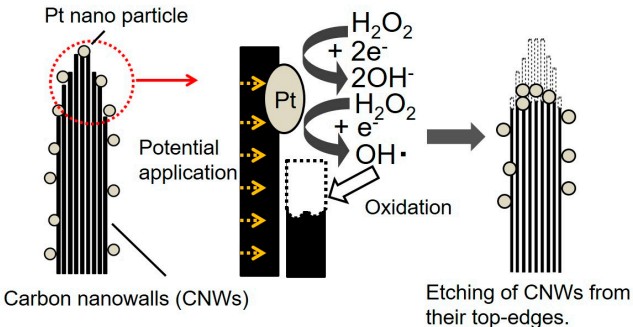

**Figure 6.** Illustration of degradation model of Pt-CNWs via secondary reaction with byproducts of $H_2O_2$ reduction.

## 5. Conclusions

$H_2O_2$ reactions on the surface of Pt-decorated CNWs under potential application were investigated on the platform of CNWs grown on carbon fiber paper. When cyclic tests of repeated CV measurements for 1000 cycles were performed three times in succession using Pt-decorated CNW electrodes in PBS solution with 240 μM of $H_2O_2$, the response peak currents of $H_2O_2$ reduction decreased with the number of cycles. According to the $H_2O_2$ concentration measurement using high-performance liquid chromatography, the decreased response peak currents are attributed to the decomposition of $H_2O_2$, resulting in a decrease in its concentration. After the cyclic tests for 3000 cycles, the density and height of CNWs were reduced and surface morphology changed. EDX mapping revealed that the Pt nanoparticles agglomerated around the top edges of CNWs after the cyclic test. These phenomena only occurred under potential application with $H_2O_2$. No degradation was found in either the structures or CV curves in the cases without $H_2O_2$ or potential application. It is suggested that OH radicals could induce a secondary reaction with the CNW surfaces and cause their structural degradation and decrease in the response current.

**Author Contributions:** M.T. analyzed experimental results and wrote the paper, M.H., M.H., H.K., K.I., M.S. and T.T. supervised the project.

**Funding:** This work was partly supported by the MEXT-Supported Program for the Strategic Research Foundation at Private Universities (S1511021), JSPS KAKENHI Grant Number 26286072, and the project for promoting the Research Center at Meijo University.

**Conflicts of Interest:** The authors declare no conflict of interest.

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
