# Peer review of "Electrochemical Reaction in Hydrogen Peroxide and Structural Change of Platinum Nanoparticle-Supported Carbon Nanowalls Grown Using Plasma-Enhanced Chemical Vapor Deposition"

_carbon, 2018_

Round 1
Reviewer 1 Report
Dear Authors
Your work on the structural change of Pt nanoparticles-supported carbon nanowalls during electrochemical reaction in H2O2 is very interesting. I think this work should be published. Nevertheless, I think the scientific discussion on the mechanisms scan be improved.
Authors are saying line 203 (in 3.2.) that "there is a migration and aggregation of Pt nanoparticles" during the process. This is explained later by an etching of CNW by the H2O2 by-products (such as OH).
To go further on your explanation, I think you should add a discussion based on all the recent papers on catalytic etching of carbon materials (such as CNW). Because my point of view is that you have a catalytic etching effect here of your carbon material. See for example the recent paper from K. Yoshida et al "Catalytic Etching of Multi-Walled Carbon Nanotubes Controlled by Oxygen Gas Pressure" published in ChemCatChem 2018, 10, 2205 – 2209, and all other papers cited in references.
Author Response
Dear editor and reviewers,
We sincerely appreciate your careful reviewing and useful comments. According to your suggestions and advices, we revised our manuscript as follows. We believe this is appropriately revised for publication.
Reviewer 1
Comments and Suggestions
Authors are saying line 203 (in 3.2.) that "there is a migration and aggregation of Pt nanoparticles" during the process. This is explained later by an etching of CNW by the H2O2 by-products (such as OH). To go further on your explanation, I think you should add a discussion based on all the recent papers on catalytic etching of carbon materials (such as CNW). Because my point of view is that you have a catalytic etching effect here of your carbon material. See for example the recent paper from K. Yoshida et al "Catalytic Etching of Multi-Walled Carbon Nanotubes Controlled by Oxygen Gas Pressure" published in ChemCatChem 2018, 10, 2205 – 2209, and all other papers cited in references.
Reply to the Reviewer 1’s comments
We sincerely appreciate your useful comments. As you pointed out, catalytic effect is also considered and important. According to your comments, we added the following sentence at lines 330-331 and cited the recommended paper (K. Yoshida et al, ChemCatChem 10, 2205–2209 (2018)) as reference [37] in the discussion section (section 4).
“Furthermore, it was also known that carbon materials such as CNTs in contact with catalytic Pt nanoparticles were etched in oxygen ambient [37].”

Reviewer 2 Report
This paper describes the fabrication and characterisation of a carbon nanowall (CNW) structure with application as an electrode for H2O2 detection.
The paper could benefit from proof reading to improve the standard of English.
In line 152, a cleaning process is mentioned. More detail of this process should be included.
It would be helpful to include voltammetry curves from the CNWs without Pt nanoparticles for comparison. Lines 225-226 describe these curves but they are not shown in the paper.
In table 1, there is a drastic increase in the ID/IG and ID'/IG ratios for the samples immersed in both H2O2 and PBS without voltammetry (column iv and v). The authors should double-check and comment on this as it is a rather surprising result, especially as little morphological change was seen in the SEM images in figure 4. Looking at the curves themselves in figure 5, I would expect the ID/IG ratio to be closer to 2.3.
Author Response
Dear editor and reviewers,
We sincerely appreciate your careful reviewing and useful comments. According to your suggestions and advices, we revised our manuscript as follows. We believe this is appropriately revised for publication.
Reviewer 2
Comments and Suggestions (1)
The paper could benefit from proof reading to improve the standard of English.
Reply to Comments and Suggestions (1)
Thank you for a kind advice. According to the editor’s recommendation, we will request English proofreading service provided by MDPI.
Comments and Suggestions (2)
In line 152, a cleaning process is mentioned. More detail of this process should be included.
Reply to Comments and Suggestions (2)
As you pointed out, explanation about the cleaning process is not sufficient. Therefore, we add the following sentence at the lines 152.
“The CNW electrode was gently submerged in pure water and slowly rinsed for about 1 minute.”
Comments and Suggestions (3)
It would be helpful to include voltammetry curves from the CNWs without Pt nanoparticles for comparison. Lines 225-226 describe these curves but they are not shown in the paper.
Reply to Comments and Suggestions (3)
We added the voltammetry curves from the CNWs without Pt as Fig. 3 (b).
Comments and Suggestions (4)
In table 1, there is a drastic increase in the ID/IG and ID'/IG ratios for the samples immersed in both H2O2 and PBS without voltammetry (column iv and v). The authors should double-check and comment on this as it is a rather surprising result, especially as little morphological change was seen in the SEM images in figure 4. Looking at the curves themselves in figure 5, I would expect the ID/IG ratio to be closer to 2.3.
Reply to Comments and Suggestions (4)Regarding the points you pointed out, we reviewed the peak deconvolution of those Raman spectra again. As a result, ID/IG ratios of (iv) and (v) are correctly 2.38 and 2.24, respectively. Those were slightly larger than that of (i), but there is no significant difference among (iii), (iv) and (v). ID’/IG ratios were also revised. Those of (iv) and (v) are 0.13 correctly. Based on these corrected values, we revised Table. 1. And, according to this result, one possible reason of increase in ID/IG ratios by (iii) 3000 cycle with PBS w/o H2O2, (iv) immersion in PBS with H2O2, and (iv) immersion in PBS w/o H2O2 is well know intercalation of ions between graphene layers, because the PBS solution includes light metals, such as sodium and potassium. The increases of ID by the intercalation have been known in the cases of CNT and graphene, for example, in the previous papers (references [29] and [30] in the revised manuscript). In these revisions, there is no change in our consideration and conclusion.
About these, we revised the following 3 points.
1. ID/IG and ID’/IG values in Table. 1.
2. Add the following description at lines 303-307.
“On the other hand, ID/IG ratio increased by 3000 cycle with PBS without H2O2. In addition, those also increased by immersion in PBS with and without H2O2. There is no significant deference whether the potential was applied or not. One possible reason of such the increases in ID/IG ratios is well known intercalation of ions between graphene layers, because the PBS solution includes light metals, such as sodium and potassium [29-30].”
3. Add the following references.
[29] Cano-Marquez, A.G.; Rodriguez-Macias, F.J.; Campos-Delgado, J.; Espinosa-Gonzalez, C.G.; Tristan-Lopez, F.; Ramirez-Gonzalez, D.; Cullen, D.A.; Smith, D.J.; Terrones, M.; Vega-Cantu, Y.I. Ex-MWNTs: Graphene Sheets and Ribbons Produced by Lithium Intercalation and Exfoliation of Carbon Nanotubes. Nano Lett. 2009, 9, 1527–1533, doi:10.1021/nl803585s.
[30] Bay, C.W.; Kong, H. Electrochemical Synthesis and Characterization of Ferric Chloride-Graphite Intercalation. Science 1998, 36, 383-390,

Round 2
Reviewer 1 Report
Your revised paper can now be published.